# The Ratio of RAC1B to RAC1 Expression in Breast Cancer Cell Lines as a Determinant of Epithelial/Mesenchymal Differentiation and Migratory Potential

**DOI:** 10.3390/cells10020351

**Published:** 2021-02-08

**Authors:** Caroline Eiden, Hendrik Ungefroren

**Affiliations:** 1First Department of Medicine, University Hospital Schleswig-Holstein, Campus Lübeck, D-23538 Lübeck, Germany; caroline.eiden@student.uni-luebeck.de; 2Clinic for General Surgery, Visceral, Thoracic, Transplantation and Pediatric Surgery, University Hospital Schleswig-Holstein, Campus Kiel, D-24105 Kiel, Germany

**Keywords:** breast cancer, epithelial subtype, mesenchymal, phenotype, chemokinesis, migration, invasion, RAC1, RAC1B, diagnostic marker

## Abstract

Breast cancer (BC) is a heterogenous disease encompassing tumors with different histomorphological phenotypes and transcriptionally defined subtypes. However, the non-mutational/epigenetic alterations that are associated with or causally involved in phenotype diversity or conversion remain to be elucidated. Data from the pancreatic cancer model have shown that the small GTPase RAC1 and its alternatively spliced isoform, RAC1B, antagonistically control epithelial–mesenchymal transition and cell motility induced by transforming growth factor β. Using a battery of established BC cell lines with either a well-differentiated epithelial or poorly differentiated mesenchymal phenotype, we observed subtype-specific protein expression of RAC1B and RAC1. While epithelial BC lines were RAC1B^high^ and RAC1^low^, mesenchymal lines exhibited the reverse expression pattern. High RAC1B and/or low RAC1 abundance also correlated closely with a poor invasion potential, and vice versa, as revealed by measuring random cell migration (chemokinesis), the preferred mode of cellular movement in cells that have undergone mesenchymal transdifferentiation. We propose that a high RAC1B:RAC1 ratio in BC cells is predictive of an epithelial phenotype, while low RAC1B along with high RAC1 is a distinguishing feature of the mesenchymal state. The combined quantitative assessment of RAC1B and RAC1 in tumor biopsies of BC patients may represent a novel diagnostic tool for probing molecular subtype and eventually predict malignant potential of breast tumors.

## 1. Introduction

Breast cancer (BC) is a heterogeneous disease with at least twenty histological subtypes of invasive breast cancer, based on growth patterns and morphology, and three biological subtypes, characterized by the expression of estrogen receptor (ER) and progesterone receptor (PR), and human epidermal growth factor receptor 2 (HER2), or the lack of all three receptors (triple-negative BC (TNBC)). The “omics“ technologies have confirmed and further stratified molecular heterogeneity of this disease into biologically and clinically meaningful subtypes, including at least six intrinsic subtypes (normal, claudin-low, luminal A and B, HER2-enriched, and basal) and four TNBC molecular subtypes (basal-like 1 and 2, mesenchymal, and luminal androgen receptor) [1]. TNBC is among the most aggressive and early metastasizing tumors [2,3], and the poor treatment response and early therapeutic escape from conventional treatments eventually leads to aggressive metastatic disease. The heterogeneity underlying BC and, particularly, tumors from TNBC patients, renders therapeutic strategies based on one-size-fits-all approach ineffective. Rather, more personalized therapies are needed that take into account the biological characteristics of each patients’ tumor, but this requires suitable prognostic biomarkers for improving diagnosis or risk assessment, and predictive biomarkers for guiding selection of optimal therapeutic interventions.

Central to the development of malignant breast cancer subtypes is the process of epithelial–mesenchymal transition (EMT) that endows cancer cells with the ability for invasion and metastasis, cancer stem cell generation, and drug resistance [4,5]. Recently, a new classification of cancer cells according to the EMT state, partial/hybrid EMT (p-EMT), or complete EMT (c-EMT) has been suggested. C-EMT is based on a transcriptional program, i.e., downregulation of E-cadherin at the mRNA level, while most tumors lose their epithelial phenotype through an alternative program involving protein internalization rather than transcriptional repression, resulting in a p-EMT phenotype [6]. The type of EMT also determines the mode of cell migration. While cells predicted to be c-EMT cells migrate as single cells in a random/spontaneous manner, p-EMT cells prefer to move as clusters and in a directed fashion [6]. Of note, the histomorphologically determined differentiation state strongly overlaps with the predicted EMT type, in the sense that cell lines with epithelial differentiation (i.e., MCF-7 and MDA-MB-468) have been predicted p-EMT cells, while MDA-MB-231 cells represent c-EMT cells [6].

BC shares in common with pancreatic ductal adenocarcinoma (PDAC) a couple of similarities, i.e., the desmoplastic stroma and the marked subtype diversification into epithelial/classical/luminal, quasi-mesenchymal/squamous, and basal-like subtypes [7]. However, the molecular drivers and, in particular, the non-mutational/epigenetic alterations responsible for phenotype diversity or conversion remain to be elucidated. In a panel of established PDAC-derived cell lines, we have recently identified high expression of RAC1B, an alternatively spliced isoform of the small GTPase, RAC1, to be closely associated with a well-differentiated phenotype and low invasive potential, while poorly differentiated cell lines of high invasive capacity presented with little or no RAC1B protein [8,9]. In addition, RAC1B promoted the expression of epithelial genes (i.e., *CDH1*) and inhibited that of mesenchymal and invasion-associated genes (i.e., *VIM* and *SNAI2*) [8], enhanced tumor-suppressive Smad signaling and inhibited tumor-promoting MEK-ERK signaling [8,10], and inhibited transforming growth factor β1 (TGFβ1)-induced EMT and cell invasion in both PDAC [9,10] and TNBC [11] cells. Of note, BC was among the first cancers reported to express RAC1B [12]. Given the similarities between pancreatic and breast cancer, we reasoned that the differential expression pattern of RAC1B in epithelial and mesenchymal-type PDAC cells should also be found among BC cells, or in other words, the subtype identity should be reflected in the expression pattern of RAC1B. We, therefore, screened a panel of established BC cell lines of known subtype/EMT type for protein expression levels of RAC1B and RAC1 as well as for migratory/invasive abilities in vitro.

## 2. Materials and Methods

### 2.1. Cell Lines

A total of eight breast cancer cell lines with different tumor origin and transcriptional subtype were employed: five invasive ductal carcinomas (IDCs)—BT-20 (basal A), MCF-7 (luminal), MDA-MB-435s (basal B), T-47D (luminal), and ZR-75-1 (luminal)—and three adenocarcinomas—MDA-MB-231 (basal B), MDA-MB-468 (basal A), and SKBR-3 (luminal). MCF-7, T-47D, and ZR-75-1 cells are luminal A-type and ER and PR-positive, whereas SKBR-3 cells are HER2-positive [13]. Among the three TNBC lines (BT-20, MDA-MB-231, and MDA-MB-468), the latter are the least metastatic [14]. All cell lines, except T-47D and SKBR-3, were maintained in RPMI-1640 medium supplemented with 10% fetal bovine serum (FBS) and 1% penicillin/streptomycin (P/S). T-47D cells were cultured in DMEM without phenol red, supplemented with 10% FBS, 1% P/S, and 10 ng/mL insulin. SKBR-3 cells were cultured in McCoy’s 5A medium with 10% FBS and 1% P/S. MCF-7, T-47D, MDA-MB-468, SKBR-3, and ZR-75-1 cells exhibit a typical epithelial morphology in 2D culture, whereas MDA-MB-435s and MDA-MB-231 are mesenchymal with a spindle-shaped morphology. The differentiation phenotype of BT-20 cells has been poorly characterized so far. These cells exhibit a mesenchymal morphology when grown on tissue culture plastics, however, they were reported to express E-cadherin [15] and to form only few and very small colonies in soft agar [16].

### 2.2. Quantitative Real-Time RT-PCR

The procedure of quantitative real-time RT-PCR (qPCR) and the primer sequences used for amplification of RAC1B, RAC1, and GAPDH as an internal control, were described in detail elsewhere [8,9,10]. The Ct values for RAC1B and RAC1 were normalized to those of GAPDH in the same sample to account for small differences in cDNA input.

### 2.3. Immunoblotting

Cell lysis and immunoblotting were essentially performed as described previously [8,9,10] with minor modifications. Proteins were fractionated by polyacrylamide gel electrophoresis on TGX Stain-Free FastCast gels (BioRad, Munich, Germany). Following blotting and antibody treatment, chemoluminescent detection of proteins was carried with a ChemiDoc XRS+ System with Image Lab Software (BioRad) with Amersham ECL Prime Detection Reagent (GE Healthcare, Munich, Germany). The signals for the proteins of interest were normalized to either the total amount of protein in the same lane (when using the TGX Stain-Free FastCast gels), or to bands for the housekeeping gene GAPDH. Significant differences (*p* < 0.05)-calculated with the unpaired two-tailed Student’s *t* test–denoted by bars above the respective cell lines. The antibodies used were Rac1b (#09-271, Merck Millipore, Darmstadt, Germany), E-cadherin (#610181) and Rac1 (#610650) (BD Transduction Laboratories, Heidelberg, Germany), GAPDH (14C10, #2118, Cell Signaling Technology, Frankfurt am Main, Germany), Vimentin (clone V9, #V6630, Sigma-Aldrich, Steinheim, Germany), and HA (clone 12CA5, Roche Diagnostics, Mannheim, Germany).

### 2.4. Chemokinesis Assays

We employed the xCELLigence^®^ DP system (ACEA Biosciences, San Diego, CA, USA) to measure random/spontaneous cell migration (also termed chemokinesis) according to previous descriptions [9,10,11]. Each well of the CIM plates received 60,000 cells in the respective growth medium of the cell lines supplemented with 1% FBS. Data acquisition was done at intervals of 15 min and the assays were run for 8 or 24 h and analyzed with RTCA software, version 1.2 (ACEA Biosciences). Prior to some migration assays, MDA-MB-468 or MDA-MB-435s cells were transfected with either RAC1B or RAC1 siRNA, or expression vectors for RAC1 mutants, RAC1-L61 or RAC1-N17 (both in the pRK5myc vector), as outlined in detail elsewhere [8,9,10]. MDA-MB-231 cells were stably transfected with a hemagglutinin (HA)-tagged version of human RAC1B in the pCGN vector followed by selection with hygromycin B (Sigma, Deisenhofen, Germany) as described previously for the PDAC-derived cell line, Panc1 [17]. BT-20 cells were treated during the assay with the RAC1 inhibitor NSC23766 (Calbiochem, Darmstadt, Germany).

## 3. Results

### 3.1. RAC1B Expression Is Associated with Differentiation and Invasive Potential in a Series of Breast Carcinoma Cell Lines

BC cell lines with two different phenotypes, i.e., epithelial (MCF-7, MDA-MB-468, T-47D) and mesenchymal (MDA-MB-231, MDA-MB-435s) as well as a cell line with an as yet poorly characterized phenotype (BT-20) were employed for this study. Initially, the phenotypic classification was confirmed by immunoblotting of the epithelial marker E-cadherin and the mesenchymal marker vimentin, as it is well documented that loss of E-cadherin expression and concomitant increases in vimentin are signs of an EMT and hallmarks of BC progression. We observed that MCF-7, T-47D, and MDA-MB-468 cells were positive for E-cadherin and negative for vimentin, while the reverse was true for MDA-MB-231 and -435s cells (Appendix A). Our results are in agreement with those from earlier studies demonstrating E-cadherin expression in MCF-7, MDA-MB-468, and T-47D but not MDA-MB-231 cells, and vimentin expression in MDA-MB-231 cells [18,19,20]. Of note, despite their mesenchymal morphology, BT-20 cells resembled epithelial-like cells with respect to expression of these markers (Appendix A).

The six cell lines were then analyzed for expression of RAC1B and RAC1 by immunoblotting (Figure 1A). MCF-7, T-47D, MDA-MB-468, and BT-20 cells abundantly expressed RAC1B, while the RAC1B protein levels in the two mesenchymal lines were significantly lower (MDA-MB-435s) or almost undetectable (MDA-MB-231) (Figure 1B). Conversely, protein levels of the parental RAC1 isoform were lower in the epithelial cell lines and were strongly upregulated in their mesenchymal-subtype counterparts with the highest levels seen in MDA-MB-435s and -231 cells (Figure 1B). A similar pattern was obtained for the steady-state mRNA levels of RAC1B and RAC1, except that those in T-47D cells were the highest in this panel, while those in MDA-MB-231 cells, unlike the protein data, were not higher than in the other cell types (Appendix A). The reciprocal expression pattern of RAC1B and RAC1 became even more apparent when their protein or mRNA expression scores were displayed as ratios of RAC1B to RAC1 (Figure 1B and Appendix A). Compared to MCF-7 cells this ratio was ~5-fold lower in MDA-MB-435s and more than 100-fold lower in MDA-MB-231 cells (Figure 1B). Although additional cell lines need to be tested, the data nevertheless suggest that the ratio of RAC1B to RAC1 grossly reflects the state of differentiation or EMT phenotype of a given BC cell.

### 3.2. RAC1B and RAC1 Expression in BC Cell Lines Is Negatively and Positively, Respectively, Associated with Migratory Activity

Given the finding of high RAC1B and low RAC1 expression in well-differentiated BC cell lines, and vice versa, we addressed the question of whether the ratio of both isoforms is predictive of their invasive potential. Using impedance-based xCELLigence technology, we monitored in real-time the rate of random migratory activity (chemokinesis) as this is associated with—and therefore a good readout for—the mesenchymal/c-EMT phenotype. As strong constitutive chemokinetic activity of both MDA-MB-231 and -435s cells has already been demonstrated in a previous study [11], we used these cells as positive controls. MCF-7, T-47D, and MDA-MB-468 cells, and two additional well-differentiated luminal A lines, SKBR-3 and ZR-75-1, exhibited no or only a low invasive potential, while the highly undifferentiated/c-EMT cell lines, MDA-MB-231 and MDA-MB-435s, were highly migratory (Figure 2A). These data are consistent with MCF-7 and T-47D cells being poorly and MDA-MB-231 and -435s cells being highly invasive/metastatic; thus, representing the opposite extremes of BC progression [21]. BT-20 cells stood out here in that their high migratory activity contrasts with their luminal A origin and epithelial phenotype.

To analyze if this disparate migration behavior is a function of RAC1B expression, we either knocked down RAC1B in MDA-MB-468 cells (RAC1B^high^) by RNA interference (Figure 2B, left-hand graph) or ectopically expressed a HA-tagged version of human RAC1B in MDA-MB-231 cells (RAC1B^low^) (Figure 2B, right-hand graph) and, intriguingly, observed an increase or decrease, respectively, in chemokinesis. We then assessed the effects of inhibiting the parental isoform, RAC1, in MDA-MB-435s cells (RAC1B^low^) by RNA interference. Knocking down RAC1 strongly decreased their chemokinetic activity compared to control cells (Figure 2C, left-hand graph). Likewise, treating BT-20 cells with various concentrations of the chemical RAC1 inhibitor, NSC23766, inhibited migration in a dose-dependent manner (Figure 2C, right-hand graph). Finally, we sought to evaluate the consequences of modulating RAC1 activity using ectopic expression of RAC1^T17N^ (RAC1-N17, GTP binding-deficient) or RAC1^Q61L^ (RAC1-L61, constitutively active) in MDA-MB-435s or MDA-MB-468 cells, respectively. Rendering RAC1 inactive clearly repressed random cell migration in MDA-MB-435s cells in a dominant-negative fashion (Figure 2D, left-hand graph), whereas turning RAC1 constitutively active enhanced chemokinesis in MDA-MB-468 cells (Figure 2D, right-hand graph). Given the antagonistic function of RAC1B and RAC1 on cell migration, we conclude that the RAC1B:RAC1 ratio determines the net migratory activity and, therefore, may be used to predict the invasive potential of newly established primary BC cell lines.

## 4. Discussion

We have previously shown that knockdown of exon 3b of *RAC1*, which specifically prevents the generation of RAC1B but not RAC1 protein, enhanced spontaneous migratory activity of MDA-MB-231 and MDA-MB-435s cells [11], although the effect was comparatively small due to low basal expression of RAC1B in these cell lines (see Figure 1). Here, we have extended our analysis on the role of these GTPases in BC by comparing cell lines with epithelial and mesenchymal phenotypes with respect to their expression levels and functional association with invasive abilities. As previously observed in a comparable set of PDAC-derived cell lines [8,9], RAC1B expression was clearly associated with an epithelial phenotype and protein levels successively decreased as cells become less differentiated and more invasive, consistent with positive and negative regulation, respectively, of *CDH1* and *VIM* by RAC1B [8]. However, unlike the situation in PDAC where protein and mRNA levels of RAC1 remained fairly constant among cell lines, the abundance of RAC1 in BC was dramatically increased in the most undifferentiated and invasive cell line, MDA-MB-231. This resulted in a reciprocal/inverse expression pattern of both RAC1 isoforms along the transition from the epithelial to the mesenchymal state. Intriguingly, the expression ratios of RAC1B:RAC1, except for BT-20 cells, were predictive of the cells’ invasive potential. For instance, the RAC1B:RAC1 ratio in MDA-MB-468 cells was higher than those in MDA-MB-231 or -435s cells, which is in agreement with reports that MDA-MB-468 cells are less metastatic than MDA-MB-231 or -435s cells [14]. Of note, BT-20 cells showed a high RAC1B:RAC1 ratio and expressed E-cadherin (Appendix A and the work in [15]) but no vimentin, consistent with the notion that basal A cells in contrast to basal B cells lack vimentin expression [13]. However, despite these epithelial characteristics, BT-20 cells exhibited a mesenchymal morphology and high basal motility in xCELLigence assays. In contrast, another cell line of the basal A transcriptional subtype, MDA-MB-468, lacked chemokinetic activity in xCELLigence assays, but knockdown of RAC1B, or ectopic expression of RAC1-L61, rendered them migration-active. Conversely, enforced expression of RAC1B or RAC1-N17 in MDA-MB-231 or -468 cells, respectively, inhibition of RAC1 by RNAi in MDA-MB-435s cells, or by NSC23766 in BT-20 cells, all decreased their chemokinetic activities (Figure 2).

These data provide functional proof of the roles of RAC1B and RAC1 as an invasion inhibitor and promoter, respectively, and are in support of a tumor suppressor function of RAC1B in both PDAC and BC. In these tumor entities, RAC1B is also a prominent example for a splice product that is tumor-protective as opposed to the parental isoform, RAC1 [22]. Our results contrast earlier findings in SCp2 mouse mammary epithelial cells, where RAC1B’s functions were more compatible with those of a promoter of EMT and invasion [23]. Species-specific differences and/or the non-tumorigenic nature of this cell line may account for this discrepancy.

We have determined here the relative expression levels of the RAC1B and RAC1 proteins among different BC cell lines; however, the ultimate biological activity of both isoforms is also determined by their GTPase activity. The constitutive activity of RAC1B resulting from the inclusion of exon 3b [24,25] might partially compensate for its generally lower expression compared to RAC1. In a previous study, we noted only moderate differences in the activity/expression ratios for RAC1B, suggesting that the amount of active RAC1B in a given cell is mainly regulated by expression of the protein rather than by a modified activity [17]. A high RAC1B:RAC1 ratio may be required for maintaining epithelial differentiation and/or blocking mesenchymal conversion and therefore a malignant phenotype. From this it follows that cell-intrinsic or environmental cues that are able to alter this ratio are believed to impact on differentiation state and eventually malignant conversion. For instance, downregulation of RAC1B may increase the cells’ sensitivity to EMT induction by stromal cell-derived TGFβ and if persisting may permit generation of a stable mesenchymal/EMT phenotype. The ability of RAC1B to counteract TGFβ-induced EMT [10] supports this assumption. We have preliminary evidence to indicate that chronic exposure of PDAC cells to TGFβ1 in vitro can change the RAC1B:RAC1 ratio by selective downregulation of RAC1B (H.U., unpublished observation). Eventually, the resulting preponderance of RAC1 is further enhanced by the ability of TGFβ to induce RAC1 GTP loading/activation (reviewed in [22]).

Of particular interest to clinicians are basal-like carcinomas, due to their poor prognosis, higher incidence in younger women, and resistance to endocrine therapy for hormone receptor-positive or trastuzumab for HER2-positive forms of the disease. As histology is not consistently distinctive, the identification of novel clinically practical biomarkers that act as surrogate measures of subtype identity or disease states is urgently needed to improve the accuracy of diagnosis or risk assessment (via prognostic biomarkers) and to inform the clinician on the optimal therapeutic strategy (via predictive biomarkers) [26]. Based on the results from this study, we recommend to determine the relative expression of RAC1B and RAC1 in clinical tumor biopsy samples, i.e., by immunohistochemistry, and evaluate their diagnostic value for assessing molecular subtype or predicting invasive potential. From a therapeutic perspective, increasing the RAC1B:RAC1 ratio by stimulating the generation of RAC1B over RAC1 through intervention with the splicing process [27,28] may be a feasible approach to reverse mesenchymal differentiation and eventually reduce metastasis or restore drug-sensitivity in BC.

## Figures and Tables

**Figure 1 cells-10-00351-f001:**
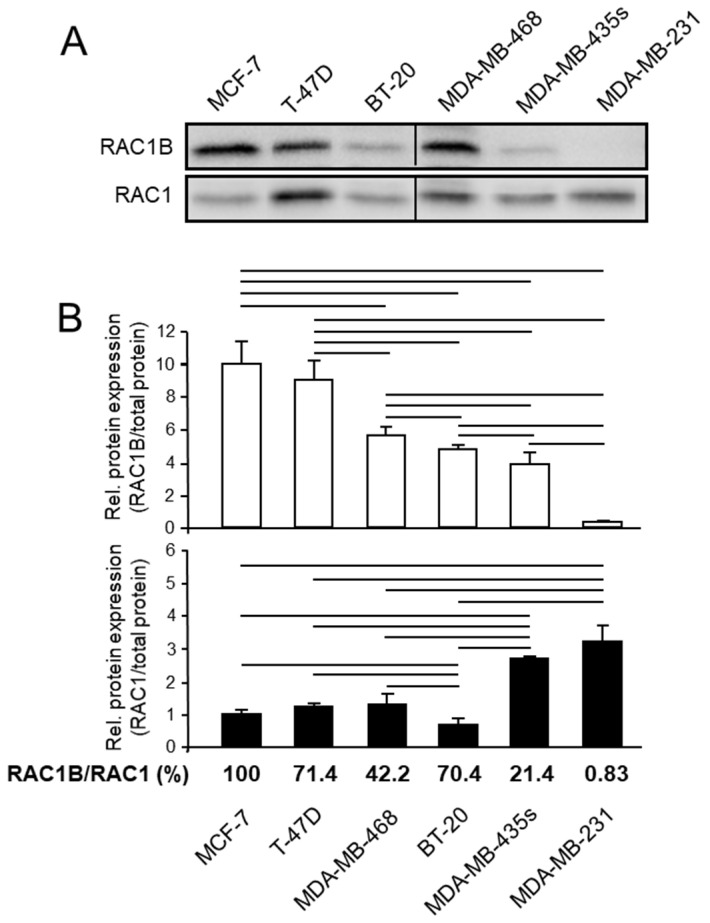
RAC1B and RAC1 expression in various established BC cell lines of either an epithelial or a mesenchymal phenotype. (**A**) Cells were grown to ~80% confluence before preparation of cell lysates and processing for immunoblotting with antibodies to RAC1B (upper blot) or RAC1 (lower blot). Shown is a representative blot. (**B**) The graphs show results from densitometric readings of RAC1B and RAC1 band intensities after normalization to the amount of total protein in the same lane. Data represent the combined results of three samples for each cell line taken at different times during continuous culture (mean ± SD, *n* = 3). The order of the cell lines was chosen according to their RAC1B protein content. The bars above the columns indicate significant differences (*p* < 0.05). The ratios of RAC1B to RAC1 (RAC1B/RAC1) in the various cell lines are given below the graphs relative to that in MCF-7 cells set arbitrarily at 100%.

**Figure 2 cells-10-00351-f002:**
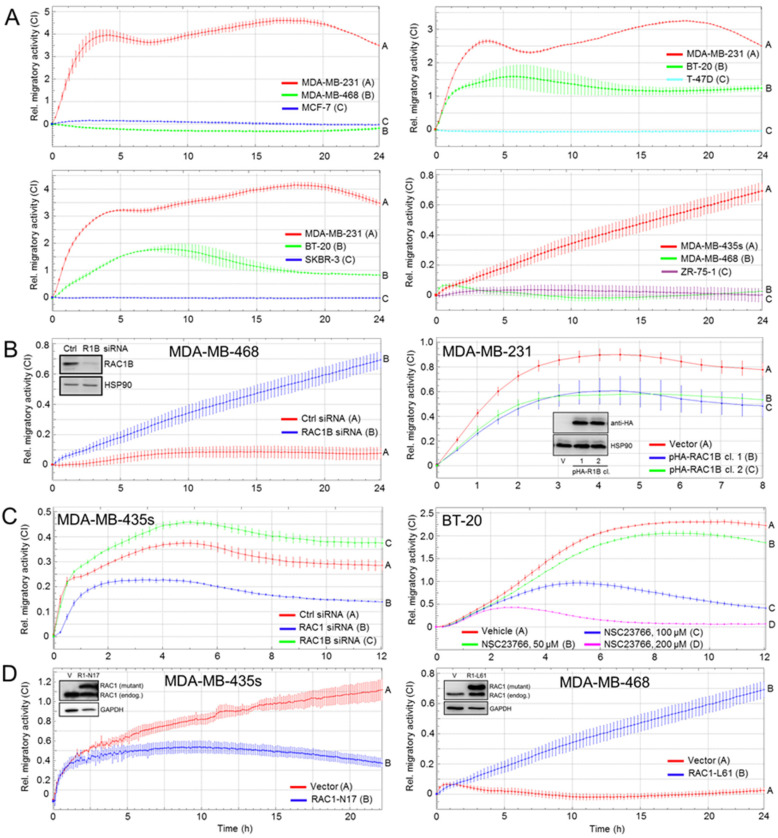
Real-time measurement of cell migration in established BC cell lines representing different histomorphological and transcriptional subtypes. (**A**) The indicated cell lines were subjected to real-time measurement of cell migration (chemokinesis setup) on an xCELLigence platform. Each graph shows the mean ± SD of 3–4 wells per cell line and is representative of three assays. (**B**) Left-hand graph, MDA-MB-468 cells were transfected with irrelevant control (Ctrl) siRNA or siRNA directed against exon 3b of *RAC1* (R1B). Successful knockdown of RAC1B was verified by immunoblotting of RAC1B with HSP90 as a loading control (inset). Right-hand graph, MDA-MB-231 cells stably expressing either empty vector (V) or HA-tagged RAC1B (two individual clones (cl.), verified with an anti-HA antibody, clone 12CA5, Roche, inset) were subjected to xCELLigence assay. Data (mean ± SD of 4 wells) are from a representative assay. (**C**) Left-hand graph, MDA-MB-435s were transfected with Ctrl siRNA, RAC1 siRNA, or RAC1B siRNA as Ctrl as outlined in subfigure (**B**). Successful knockdown of RAC1 was verified by immunoblotting (not shown). Right-hand graph, BT-20 cells were left untreated or treated with the indicated concentrations of NSC23766 during the migration assay. (**D**) MDA-MB-435s (left-hand graph) or MDA-MB-468 (right-hand graph) cells were transfected with empty vector and either RAC1-N17 or RAC1-L61, respectively, and 48 h later subjected to measurement of migratory activities. Data are the mean ± SD of 4 wells and are representative of three assays. Overexpression of the RAC1 mutants was verified by RAC1 immunoblotting (insets); endog., endogenous.

## Data Availability

Not applicable.

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
