# Peer review of "The Ratio of RAC1B to RAC1 Expression in Breast Cancer Cell Lines as a Determinant of Epithelial/Mesenchymal Differentiation and Migratory Potential"

_cells, 2021, doi:10.3390/cells10020351_

Round 1
Reviewer 1 Report
Although the topic is highly interesting and hot, I found the presented results very much preliminary and few. The data on cell lines are correct but there are only two types of experiments (Western blot and migration test) supporting the possible significance of the author's result. More experiments are needed to make especially in in vivo system. It would have been more valuable and promising if the author did some measurements on various cancer biopsy (epitheloid, mesenchymal-like invading, metastatic cancer cells) to prove the relevancy of this result. I agree that to make fine discrimination between various type of breast cancer, to make early diagnosis, to use more personal therapy is urgent and very much important for the therapy. I accept that looking at the RAC1B:RAC1 ratio can provide a good molecular diagnostic prognosis, but the author should support this with more results.
Author Response
Dear Editor, dear Matin,
Please, find enclosed the revised version of our manuscript (Manuscript ID: cells-1061170).
We thank the reviewers for their generally positive comments on this work and have done our best to satisfy their concerns. We are particularly thankful to reviewers 2 and 3 since their requests for confirmation of the proposed differentiation phenotype of the various cell lines by EMT marker protein analysis has greatly aided in solving a contradiction/confusion regarding the BT-20 cell line.
We believe that we have found a good compromise in balancing the requests of the reviewers with the “Communications” format that we have chosen because, originally, we could not provide enough data for a regular paper. Unfortunately, due to the COVID-19 pandemics we were unable to complement this study with data from primary tissues or cell lines (specifically requested by reviewer 1), or in vivo experiments (specifically requested by reviewer 3) due to shutdown of our animal facility.
All changes have been highlighted by the “track changes” mode for easy visualization. We are confident that the revisions have greatly improved the quality of this manuscript and are looking forward to its final acceptance in “Cells”.
Reviewer 1
Although the topic is highly interesting and hot, I found the presented results very much preliminary and few. The data on cell lines are correct but there are only two types of experiments (Western blot and migration test) supporting the possible significance of the author's result. More experiments are needed to make especially in in vivo system.
It would have been more valuable and promising if the author did some measurements on various cancer biopsy (epitheloid, mesenchymal-like invading, metastatic cancer cells) to prove the relevancy of this result.
Response: We totally agree with this point and analysis of tissue specimen and biopsies is crucial to confirm our results. However, unfortunately, we do not have readily access to primary tissues or cell lines, in part due to cancelling of many operations due to the COVID-19 pandemics. Analysis of cancer biopsies and cell lines derived thereof is scheduled for future studies.
I agree that to make fine discrimination between various type of breast cancer, to make early diagnosis, to use more personal therapy is urgent and very much important for the therapy. I accept that looking at the RAC1B:RAC1 ratio can provide a good molecular diagnostic prognosis, but the author should support this with more results.
Additional changes made:
- Eight new references have been added.
- My colleague, Caroline Eiden, has performed the immunoblots for E-cadherin and vimentin (new Figure S1) and has, therefore, been added to the author line.
Reviewer 2 Report
In this manuscript, the author determined the relative expression levels of the RAC1B and RAC1 proteins among 6 different breast cancer (BC) cell lines to test the role of RAC1B in EMT and invasiveness. This is an important issue as conflicting results have been reported in the literature on this activity. The manuscript reports two core experiments: 1) the correlation of RAC1B expression levels with the differentiation status and invasive potential of the BC cell lines, and 2) the random cell migration activity of the BC cell lines associated with the different RAC1B and RAC1 expression levels.
Although the experiments are technically rather sound and well described, I have suggestions for improvement that are required to sustain the conclusions the author wants to draw.
main suggestions:
- As referred by the author in the Discussion, lines 202-220, the comparison of the active, GTP-bound levels of RAC1B and RAC1 is an important consideration, but these experiments are missing in the characterization of the selected BC cell lines. The respective pull-down assay data should be added to support the authors conclusions and suggestions, namely that (i) active RAC1 is the main correlating factor with cell migration and EMT, (ii) that RAC1B may reduce the availability of activating stimuli (probably via GEF occupation) for RAC1, and (iii) that determination of the relative expression of RAC1B and RAC1 in clinical tumor biopsy samples could be a valuable biomarker in BC (lines 228-9). If no such correlation were revealed by the pull-down assays, then alternative mechanisms such as RAC1 or isoform-specific interaction partners should be discussed.
- a) Figure 2: the author analyzed random/spontaneous cell migration (also termed chemokinesis, line 111). It would be much more meaningful to add transwell migration or invasion assays, at least, for cell models shown in Fig. 2B.
b) I would also repeat these experiments in the presence of RAC1/RAC1B inhibitors to confirm that motility/invasion is RAC-dependent.
- a) line 141, Figure 1: The graphs show results from densitometric readings of RAC1B and RAC1 band intensities after normalization to the amount of total protein in the same lane. Although the normalization of RAC1 intensity to the total protein in the same lane is reasonable to compare its overall expression levels between different cell lines, RAC1B levels should not be normalized this way, but rather be shown as the ratio RAC1B/RAC1, as both originate from the same pre-mRNA transcript pool. Only then can the conclusion be drawn in the Discussion, line 190, that a reciprocal/inverse expression pattern of both RAC1 isoforms exists along the transition from the epithelial to the mesenchymal state.
b) also in Fig 1, epithelial (MCF-7, MDA-MB-468, T-125 47D) and mesenchymal (BT-20, MDA-MB-231, MDA-MB-435s) cell types were compared; however, it would be important to include respective marker proteins (E-cadherin, claudin, vimentin, snail or equivalent) into the shown Western blot to substantiate this classification.
- line 78: the statement: ‘Of note, BC was the first cancer reported to express RAC1B twenty years ago [12].’ is incorrect. The first report was in colorectal cancer: Oncogene 18, 6835–6839 (1999). One may say ‘was among the first cancers reported to express RAC1B’
- The landmark work on RAC1B in a breast cancer cell model by Radisky et al. (Rac1b and reactive oxygen species mediate MMP-3-induced EMT and genomic instability. Nature 436, 123–127, 2005) is not discussed. This is essential as their results argue the inverse, that RAC1B is a promotor of EMT and invasion.
minor points:
- lines 61-64: unclear sentence structure, probably a comma is missing: ‘..EMT type, in that..’ or ‘, in the sense that’ (“Of note, the histomorphologically determined differentiation state strongly overlaps with the predicted EMT type in that cell lines with epithelial differentiation (i.e., MCF-7, MDA-MB-468) have been predicted p-EMT cells, while MDA-MB-231 cells represent c-EMT cells.”)
- line 94: according to the content of this paragraph the expression ‘except T-47D’ should be ‘except T-47D and SKBR-3 cells’
- line 107: unclear procedure: ‘The signals for the proteins of interest were normalized to the total amount of protein in the same lane…’ Please specify clearly how this was done.
- line 200: this sentence provides a too general message on RAC1B and may only be true in PDAC and BC, as stated in the preceding sentence.
- page 6, line 236: Supplementary Materials: The following are available online at www.mdpi.com/xxx/s1, Figure S1: 235 title, Table S1: title, Video S1: title.
These were not available to the reviewer and are not referred to in the manuscript text. Maybe a copy error from a previous manuscript??):
Author Response
Dear Editor, dear Matin,
Please, find enclosed the revised version of our manuscript (Manuscript ID: cells-1061170).
We thank the reviewers for their generally positive comments on this work and have done our best to satisfy their concerns. We are particularly thankful to reviewers 2 and 3 since their requests for confirmation of the proposed differentiation phenotype of the various cell lines by EMT marker protein analysis has greatly aided in solving a contradiction/confusion regarding the BT-20 cell line.
We believe that we have found a good compromise in balancing the requests of the reviewers with the “Communications” format that we have chosen because, originally, we could not provide enough data for a regular paper. Unfortunately, due to the COVID-19 pandemics we were unable to complement this study with data from primary tissues or cell lines (specifically requested by reviewer 1), or in vivo experiments (specifically requested by reviewer 3) due to shutdown of our animal facility.
All changes have been highlighted by the “track changes” mode for easy visualization. We are confident that the revisions have greatly improved the quality of this manuscript and are looking forward to its final acceptance in “Cells”.
Reviewer 2
In this manuscript, the author determined the relative expression levels of the RAC1B and RAC1 proteins among 6 different breast cancer (BC) cell lines to test the role of RAC1B in EMT and invasiveness. This is an important issue as conflicting results have been reported in the literature on this activity. The manuscript reports two core experiments: 1) the correlation of RAC1B expression levels with the differentiation status and invasive potential of the BC cell lines, and 2) the random cell migration activity of the BC cell lines associated with the different RAC1B and RAC1 expression levels.
Although the experiments are technically rather sound and well described, I have suggestions for improvement that are required to sustain the conclusions the author wants to draw.
Main suggestions:
- As referred by the author in the Discussion, lines 202-220, the comparison of the active, GTP-bound levels of RAC1B and RAC1 is an important consideration, but these experiments are missing in the characterization of the selected BC cell lines. The respective pull-down assay data should be added to support the authors conclusions and suggestions, namely that (i) active RAC1 is the main correlating factor with cell migration and EMT, (ii) that RAC1B may reduce the availability of activating stimuli (probably via GEF occupation) for RAC1, and (iii) that determination of the relative expression of RAC1B and RAC1 in clinical tumor biopsy samples could be a valuable biomarker in BC (lines 228-9). If no such correlation were revealed by the pull-down assays, then alternative mechanisms such as RAC1 or isoform-specific interaction partners should be discussed.
Response: We agree with this reviewer that this is an important experiment. However, as demonstrated earlier in PDAC-derived cells, regulation of biological activity occurs at the level of protein synthesis rather than activity regulation and this likely applies also for BC cells. For the RAC1B/RAC1 ratio to be used for diagnostic purposes in the clinical practice only determination of the relative amounts of protein or mRNA levels but not the active protein fraction by pulldown assays is feasible in tumor biopsies. However, to show that (i) active RAC1 is crucial for cell migration, we have performed additional experiments with inactive and constitutively active RAC1 mutants (see new panels C and D in Figure 2). Regarding elucidation of the underlying mechanism of functional antagonism of RAC1B and RAC1 in the regulation of cell motility, reduction by RAC1B of the availability of activating stimuli for RAC1 is surely one of several possibilities. However, our previous data have shown that RAC1B knockdown led to an increase in RAC1 protein levels (Ref. 10), again pointing to alterations in protein abundance as the main driver of RAC1 function. This issue therefore requires a more thorough analysis that we believe is beyond the scope of this manuscript.
- a) Figure 2: the author analyzed random/spontaneous cell migration (also termed chemokinesis, line 111). It would be much more meaningful to add transwell migration or invasion assays, at least, for cell models shown in Fig. 2B.
b) I would also repeat these experiments in the presence of RAC1/RAC1B inhibitors to confirm that motility/invasion is RAC-dependent.
Response: The assays shown in Figure 2 are indeed all transwell assays with impedance-based measurement of those cells that have managed to pass through the 8 µm pores of the membrane separating the upper from the lower chambers. Does the reviewer mean transwell assays in a chemotaxis setup, i.e., with a serum or TGF-b gradient? We should emphasize that we have specifically chosen the chemokinesis setup because this is the preferred mode invasion by mesenchymal-type cancer cells.
- b) In order to confirm that migration is RAC1-dependent we have performed the following experiments: 1) Knockdown of RAC1 with a RAC1 siRNA or ectopic expression of inactive RAC1 (RAC1-T17N) (GTPase binding-defective, interferes with RAC1 function in a dominant-negative manner) in MDA-MB-435s cells. 2) Treatment of BT-20 cells with the specific GTP-RAC1 inhibitor NSC23766, and finally, ectopic expression in MDA-MB-468 cells of RAC1-Q61L, a constitutively active RAC1 mutant. Data from these experiments clearly show that chemokinesis is RAC1-dependent and that RAC1 is only capable of driving cell migration in its active GTP-bound form.
- a) line 141, Figure 1: The graphs show results from densitometric readings of RAC1B and RAC1 band intensities after normalization to the amount of total protein in the same lane. Although the normalization of RAC1 intensity to the total protein in the same lane is reasonable to compare its overall expression levels between different cell lines, RAC1B levels should not be normalized this way, but rather be shown as the ratio RAC1B/RAC1, as both originate from the same pre-mRNA transcript pool. Only then can the conclusion be drawn in the Discussion, line 190, that a reciprocal/inverse expression pattern of both RAC1 isoforms exists along the transition from the epithelial to the mesenchymal state.
b) also in Fig 1, epithelial (MCF-7, MDA-MB-468, T-125 47D) and mesenchymal (BT-20, MDA-MB-231, MDA-MB-435s) cell types were compared; however, it would be important to include respective marker proteins (E-cadherin, claudin, vimentin, snail or equivalent) into the shown Western blot to substantiate
Response: a) As requested, RAC1B levels are now shown, in addition, as the ratios of RAC1B/RAC1 in numerical form (in %) below the graphs in Figure 1B.
- b) This is a very good suggestion. However, this classification of epithelial and mesenchymal was not proposed by us but for all cell lines (except BT-20, see below) is established for many years based on morphological and biochemical criteria. For instance, T47D cells express much less vimentin than MDA-MB-231 cells (PMID: 29575318), MCF-7 and T47D cells exhibit higher levels of E-cadherin and lower levels of N-cadherin, fibronectin and vimentin than MDA-MB-231 cells (PMID: 29331414) and MDA-MB-468 cells express readily detectable levels of E-cadherin (by immunoblotting, PMID: 28750639, PMID: 20101232). Nevertheless, we have performed immunoblot experiments with our cell batches using antibodies to E-cadherin and vimentin. Data clearly show that E-cadherin is present and vimentin is absent in MCF-7, T-47D, MDA-MB-468 and BT-20 cells, while the reverse is true for MDA-MB-231 and MDA-MB-435s cells.
With respect to BT-20, positivity for E-cadherin and lack of vimentin expression was somewhat surprising as these cells exhibit a mesenchymal morphology when grown on tissue culture plastics. We have, therefore, carefully re-analyzed RAC1B and RAC1 expression in these cells and found that the protein levels of RAC1B had been mistakenly underestimated. This has now been corrected in Figures 1A and 1B.
The E-cadherin and vimentin immunoblots of all six cell lines have been added as Supplementary material in Figure S1.
- line 78: the statement: ‘Of note, BC was the first cancer reported to express RAC1B twenty years ago [12].’ is incorrect. The first report was in colorectal cancer: Oncogene 18, 6835–6839 (1999). One may say ‘was among the first cancers reported to express RAC1B’
Response: We apologize for this mistake. We have rephrased this statement as suggested.
- The landmark work on RAC1B in a breast cancer cell model by Radisky et al. (Rac1b and reactive oxygen species mediate MMP-3-induced EMT and genomic instability. Nature 436, 123–127, 2005) is not discussed. This is essential as their results argue the inverse, that RAC1B is a promotor of EMT and invasion.
Response: We apologize for this. We have briefly discussed this work in the Discussion section.
Minor points:
- lines 61-64: unclear sentence structure, probably a comma is missing: ‘..EMT type, in that..’ or ‘, in the sense that’ (“Of note, the histomorphologically determined differentiation state strongly overlaps with the predicted EMT type in that cell lines with epithelial differentiation (i.e., MCF-7, MDA-MB-468) have been predicted p-EMT cells, while MDA-MB-231 cells represent c-EMT cells.”)
Response: This has been rectified.
- line 94: according to the content of this paragraph the expression ‘except T-47D’ should be ‘except T-47D and SKBR-3 cells’
Response: We are sorry for this mistake. This has been corrected as suggested.
- line 107: unclear procedure: ‘The signals for the proteins of interest were normalized to the total amount of protein in the same lane…’ Please specify clearly how this was done.
Response: As requested, this procedure has now been described. It requires polyacrylamide gel electrophoresis on TGX Stain-Free FastCast gels (BioRad, Munich, Germany) and the ChemiDoc XRS+ System with Image Lab Software from BioRad.
- line 200: this sentence provides a too general message on RAC1B and may only be true in PDAC and BC, as stated in the preceding sentence.
Response: In our copy, line 200 is within the legend of Figure 1 and describes how cells were prepared for immunoblotting. Does the reviewer perhaps refer to another sentence?
- page 6, line 236: Supplementary Materials: The following are available online at www.mdpi.com/xxx/s1, Figure S1: 235 title, Table S1: title, Video S1: title.
These were not available to the reviewer and are not referred to in the manuscript text. Maybe a copy error from a previous manuscript??):
Response: This statement is part of the template and we are sorry for not having deleted it. The original version did not contain Supplementary Materials, however, the revised version now contains Figures and S1 and S2.
Additional changes made:
- Eight new references have been added.
- My colleague, Caroline Eiden, has performed the immunoblots for E-cadherin and vimentin (new Figure S1) and has, therefore, been added to the author line.
Reviewer 3 Report
The author had tried to identify the role of RAC1B and RAC1 and protein level expression in breast cancer during invasion and metastasis.
comments
- The author had investigated multiple BC cell lines to establish their claim. Comments on EMT were inconclusive as the observations were not confirmed by looking at other EMT markers.
- A 3D organoid culture will be interesting to look at. Especially, treatment with a Rac1 Inhibitor, like CAS 1177865-17-6 is essential.
- Further investigation at RNA level is essential to make that claim.
- In-vivo experiment is needed.
Author Response
Dear Editor, dear Matin,
Please, find enclosed the revised version of our manuscript (Manuscript ID: cells-1061170).
We thank the reviewers for their generally positive comments on this work and have done our best to satisfy their concerns. We are particularly thankful to reviewers 2 and 3 since their requests for confirmation of the proposed differentiation phenotype of the various cell lines by EMT marker protein analysis has greatly aided in solving a contradiction/confusion regarding the BT-20 cell line.
We believe that we have found a good compromise in balancing the requests of the reviewers with the “Communications” format that we have chosen because, originally, we could not provide enough data for a regular paper. Unfortunately, due to the COVID-19 pandemics we were unable to complement this study with data from primary tissues or cell lines (specifically requested by reviewer 1), or in vivo experiments (specifically requested by reviewer 3) due to shutdown of our animal facility.
All changes have been highlighted by the “track changes” mode for easy visualization. We are confident that the revisions have greatly improved the quality of this manuscript and are looking forward to its final acceptance in “Cells”.
Reviewer 3
The author had tried to identify the role of RAC1B and RAC1 and protein level expression in breast cancer during invasion and metastasis.
comments
- The author had investigated multiple BC cell lines to establish their claim. Comments on EMT were inconclusive as the observations were not confirmed by looking at other EMT markers.
Response: This issue was also raised by Reviewer 2. We have therefore performed Western blots with E-cadherin and vimentin antibodies and have included these data in the manuscript as Figure S1. Please, see point 3b of Reviewer 2.
- A 3D organoid culture will be interesting to look at. Especially, treatment with a Rac1 Inhibitor, like CAS 1177865-17-6 is essential.
Response: This is certainly true but this method, unfortunately, is not established in our lab yet. The use of an inhibitor was also requested by Reviewer 2 and these data are now part of Figure 2 (new panels C and D).
- Further investigation at RNA level is essential to make that claim.
Response: We have quantified RAC1B and RAC1 expression by quantitative real-time RT-PCR (qPCR) analyses and have displayed these data singly and as RAC1B:RAC1 ratios (according to a suggestion from Reviewer 2) in the new Figure S2. These data essentially confirm the respective protein data.
- In-vivo experiment is needed.
Response: Our animal facility is closed due to COVID-19 crisis. For this reason, we are unable to perform animal experiments currently.
Additional changes made:
- Eight new references have been added.
- My colleague, Caroline Eiden, has performed the immunoblots for E-cadherin and vimentin (new Figure S1) and has, therefore, been added to the author line.
Round 2
Reviewer 1 Report
The authors made corrections I accept. I agree and accept also that is difficult to get biopsy and samples from patients in this epidemic time. I hope that the authors can do detailed analysis in the future as they schedule. To increase the validity of their results and to consider the clinical application it is very much important to support their results with these data.
Author Response
We perfectly agree with the reviewers' opinion and are thankful for his understanding of the situation.
Reviewer 2 Report
In their revised manuscript, the author has addressed all essential issues in a very satisfactory manner, including the addition of requested experiments.
One minor issue remained, due to unclear line numbering:
line 200: the referred sentence is: ‘RAC1B is also a prominent example for a splice product that is tumor-protective as opposed to the parental isoform, RAC1 [15].’
This is a too general message on RAC1B and may only be true in PDAC and BC, as stated in the preceding sentence. Just adjust the wording or the connection to the preceding sentence.
Author Response
We thank the reviewer for his enthusiastic comment on the revised version.
As requested, we have fixed the minor issue by rephrasing the sentence to make clear that the statement is only true for PDAC and BC.
Reviewer 3 Report
A 3D organoid culture or an in-vivo experiment is essential to this manuscript.
Author Response
We perfectly agree with the reviewer that the addition of breast cancer tissue samples and 3D cultures would have been helpful. However, due to the reasons stated in my response to his previous comments, we are currently unable to provide these data in a reasonable amount of time mainly due to the COVID-19 situation.
We feel that our results are nevertheless interesting and sufficient for the "communications" format.
In the second revision that I have submitted on January 27, 2021, we had changed the title of our manuscript in response to a request from the Academic Editor to replace "cells" by "cell lines" and "malignant" by "migratory". These changes shifted the focus of our work to an in vitro analysis (with established cell lines rather than primary cells or patient-derived organoids, and migration assays rather than malignant behavior in vivo in animal models). As a consequence of these changes in the title, we believe that the additional experiments requested by Reviewer 3, which, unfortunately, we are unable to provide due to the COVID-19 pandemics, are not mandatory anymore.